# Numerical Simulation of Damage Evolution and Electrode Deformation of X100 Pipeline Steel during Crevice Corrosion

**DOI:** 10.3390/ma15062329

**Published:** 2022-03-21

**Authors:** Wenxian Su, Zhikuo Liu

**Affiliations:** School of Energy and Power Engineering, University of Shanghai for Science and Technology, Shanghai 200093, China; 13122995576@163.com

**Keywords:** X100 pipeline steel, electrochemical corrosion, crevice corrosion, damage evolution, electrode deformation

## Abstract

In this paper, the spatial and temporal damage evolution was described during crevice corrosion through developing a two-dimensional (2-D) model. COMSOL code was used to simulate the crevice corrosion regulated by the *I∙R* voltage of nickel (Ni) metal in sulfuric acidic. The electrode deformation, potential and current curves, and other typical characteristics were predicted during crevice corrosion, where results were consistent with published experimental results. Then, based on the Ni model, the damage evolution of X100 crevice corrosion in CO_2_ solution was simulated, assuming uniform distribution of solution inside and outside the crevice. The results showed that over time, the surface damage of Ni electrode increased under a constant applied potential. As the gap increased, the critical point of corrosion (CPC) inside the crevice moved into a deeper location, and the corrosion damage area (CDA) gradually expanded, but the threshold value of corrosion damage remained almost unchanged. The CDA inside the crevice extended toward the opening and the tip of crevice. Since the potential drop in this region increases with increasing current, the passivation potential point moved towards the opening. As the gap increased and the electrolyte resistance decreased, the critical potential for reaching the maximum corrosion rate moved into a deeper location. It is significant for predicting the initial damage location and the occurrence time of surface damage of crevice corrosion through the 2-D model that is not available through the one-dimensional simplified model.

## 1. Introduction

Characterized by high strength, high toughness, low buckling ratio, and good corrosion resistance, X100 is widely used in oil and natural gas transportation. With the development of CO_2_ mixed flooding technology (CO_2_–EOR) in oil and gas exploitation, the existence of CO_2_ gas in the pipelines is inevitable. The dissolved CO_2_ forms carbonate or bicarbonate solution, which causes corrosion of the pipelines [1,2]. In metal-insulator or metal-metal gaps, metal dissolution and limited mass-transport effect inside the crevice lead to acidification of solution internal the gap, where Cl^−^ migration keeps the solution electroneutral. With the increase of metal dissolution rate, hydrolytic reactions in the inner crevice and ionic mobility into the crevice both increase, inducing an accelerated corrosion rate. Due to the non-uniform distribution of metal ions and dissolved gases inside and outside the crevice, this leads to a potential difference (PD) in the corrosion solution to affect the kinetics of the electrode process and even establishes an electrochemical cell [3,4,5]. Corrosion behavior of thermal barrier coatings (TBCs) in molten salt at high temperature is similar with crevice corrosion. The TBCs are mainly used in critical components of aircraft gas turbine engines [6,7,8]. The corrosion of coating-molten salt-gas systems is formed at high temperatures, which is similar to the insulation–solution–metal crevice corrosion system, but the corrosion mechanism is different [9,10]. According to a large number of detailed descriptions and studies on the corrosion behavior of TBCs by Yasin, Kadir, and Abdullah et al., it can be seen that the corrosion of TBC_S_ is the cumulative effect of the chemical impurity of molten salts such as vanadium oxide (V_2_O_5_) and sodium sulfate (Na_2_SO_4_) on the coating–gas interface during 1000 °C combustion [11,12,13].

The corrosion deposition will gradually accumulate and block the crevice opening, so the crevice corrosion is generally invisible and quite dangerous [14,15,16]. Almost all metals and alloys are subject to crevice corrosion, but the sensitivity of metal to corrosive media is different. Metal and alloys with self-passivation properties (stainless steel, aluminum alloys, titanium alloys) are more susceptible to crevice corrosion in aerated neutral media with reactive anions [17,18,19]. The sensitivity of metal to crevice corrosion depends on its self-passivation ability. Some elements of Cr, Ni, Mo, N, Cu, and Si can effectively improve the corrosion resistance of stainless steel [20,21,22].

The electrode potential theory (EPT), that is, the *I∙R* theory proposed by Pickering, indicates that the first stage is O_2_ depletion followed by depolarization of Ni, causing transfer of the cathodic reaction to the exposed surface of Ni metal in solution. The PD between the exposed surface and the gap creates a coupling potential due to the depolarization effect. Because of restrained diffusion transfer, the cathodic reaction only occurs on the boldly exposed surface outside the crevice. The combination of the anode area inside the crevice and the cathode area outside the crevice forms an electrolyte current channel. Assuming that the electrolyte resistance in the flow channel is R, the current I through the channel will produce a PD of Δ*φ** = *I∙R* voltage drop inside and outside the crevice, which is the driving force that induces the transformation of metal or alloy material from passivity to activity and crevice corrosion. PD as a function of crevice depth contributes to the initiation of crevice corrosion in metals with an active-to-passive transition in Pickering’s model [23,24,25,26].

Theoretical models of crevice corrosion have been studied extensively. Abdulsalam [27] developed a model for Ni crevice corrosion and studied the effect of different potentials on corrosion rate based on Ni anodic polarization curves. Kennell [28] proposed a one-dimensional steady-state model of crevice corrosion combining the critical crevice solution theory (CCST) and the *I∙R* drop theory, which can calculate the variation of electrolyte composition concentration over time and the distribution of local corrosion current and can simulate the corresponding potential evolution according to the three processes of crevice corrosion. Xu and Song [29,30] developed a mathematical model for predicting the chemistry pipeline for variable crevices (i.e., crevice width varies with length) and explored the changes in chemical reactions and corrosion rates in pipeline surfaces with/without cathodic protection. Stenta and Kahyarian [31,32] established a one-dimensional crevice corrosion model under cathodic protection conditions to study transient electrochemical corrosion processes. Most of the aforementioned models consider only one dimension of the crevice corrosion (the effect of the crevice width on the potential and chemical parameters are ignored).

In this paper, a 2-D crevice corrosion of Ni metal in acidic electrolyte was modeled for simulating the corrosion damage evolution of Ni under different potentials and crevice widths by using COMSOL code. The initial location of corrosion damage, electrode surface deformation, interface potential distribution, and corrosion current density distribution were discerned, and the simulation and experiment were compared to provide a basis for crevice corrosion controlled by *I∙R* voltage drop. Further, based on the damage model of Ni crevice corrosion, the electrode deformation of X100 crevice corrosion in a solution containing CO_2_ was studied, and different factors on the corrosion rate were investigated.

## 2. Simulation of Ni Electrode Crevice Corrosion

The process of using the Finite Element Software COMSOL with the corrosion module to compute the crevice corrosion of Ni electrode mainly includes the following steps:

(1) Model Definition: The “corrosion, secondary interface” is used to model the problem. A 2-D model was built based on the actual gap size, consisting of a piece of metal and a piece of insulator. The depth of the gap was 10 mm, and the width was set to 0.1~0.3 mm. The change of electrolyte concentration along the width of the gap was ignored. The cell geometry was modeled by including the crevice and a 2-mm square outside the crevice mouth. An electrolyte region was used to model electrolyte charge transport.

(2) Boundary conditions: an Electrode Surface boundary node was added to model electrode surface deformation. The experimental polarization data for the electrode reaction kinetics were used as an interpolation polynomial. The potential of the electrode was set to 0.3 V. An Electrolyte Potential boundary condition was used to apply a 0 V electrolyte potential along the leftmost boundary. The default Insulation condition was used for all other boundaries. For the deformed geometry, on the nonmoving boundaries, the model was solved with the default Nondeforming Boundary. These boundaries were all straight lines, so to improve the shape of the deformation in the corners of the geometry, and also to reduce the problem size, the boundary condition setting of the feature was changed to Zero Normal Displacement.

(3) Meshing: a higher resolution mesh at the crevice mouth was built, and the maximum element size was 8 × 10^−4^ mm, the minimum element size was 3 × 10^−6^ mm, and the maximum element growth rate was 1.3. After the local mesh of keys area was refined, the maximum element size was 1.2 × 10^−4^ mm, the minimum size was 2.4 × 10^−7^ mm, and the maximum element growth rate was 1.1.

(4) Solving: the problem was solved using a Time Dependent with Initialization study. The study contains two solver steps: a Current-Initialization step first solves for the potentials only, using a stationary solver. The second Time-Dependent step solves for the full problem for the prescribed 50 h duration. The change curve of electrolyte potential and deformation geometry was plotted; the corrosion current density distribution along the electrode surface at different times was plotted; and the potential difference distribution between the potential in the electrode and the electrolyte potential along the nickel surface was plotted.

(5) Convergence test: We refined the meshes of the key areas of the model and compared the calculation results before and after refinement. When the results did not change substantially but converged to a certain value, we considered the mesh to have converged and no further refinement was required.

### 2.1. Geometric Model

A 2-D model of the crevice consisting of epoxy resin (insulator) to contact with Ni metal vertically is presented in Figure 1. One side of the Ni metal is in a narrow channel, and the other is exposed to the electrolyte, forming an occlusion region. The internal electrolyte of the crevice, marked as Part I, is oriented with the origin at the crevice opening. The *x*-direction is positive from the opening toward the crevice tip. The *y*-positive direction is from the crevice bottom upwards. Part II defines the external electrolyte region. In Part I, the width of the crevice is defined by *W*, and the depth along the *x*-direction is defined by *L*, respectively. In Part II, the width of the electrolyte region outside is denoted by *M* and the width of the exposed surface area is denoted by *N*.

### 2.2. Potential Boundary Conditions

The polarization curve for a planar Ni electrode is shown in Figure 2. The polarization curve includes two regions: one is the active region, where the current density increases with the increased anodic polarization, and the other is the passive region, where the current density decreases or remains constant with the increased polarization. The electrolyte of pH = 0.3 and 0.5 mol/L H_2_SO_4_ at 24 °C was used to model the electrolyte charge transport, with a conductivity of *k* = 0.184 S/cm, assuming that the properties (concentration, pH value, conductivity, etc.) of the electrolyte inside the crevice remained constant. Ni metal surface contacted the electrolyte solution as an anode. The corrosion current density was expected to be adequately small to ignore the convective mass transfer and electrolyte solution diffusion effect; only the mass transfer of electromigration and the current generated by ion electromigration were considered. As the anode current flowed from inside the gap through the electrolyte to the uncovered metal surface, an ohmic voltage drop occurred along the length of the gap. The magnitude of this potential drop at the transition location from active potential to passive potential is given by Ohm’s law, Δ*φ** = *I∙R*, when the transition occurs between the active state and the passive state of the Ni surface, where *I* is equal to *i_crit_* multiplied by the surface area of the crevice, *i_crit_* is the critical current density when the transition occurs between the active state and the passive state, and *R* is the resistance of the electrolyte solution inside the crevice, given by Formula (1):(1)$$R=\frac{\rho \cdot x}{A}$$
where, *ρ* denotes the electrolyte resistivity inside the crevice. *x* is the location along the crevice depth. *A* is the cross-sectional area of the crevice opening.

The potential boundary conditions (PBCs) of this 2-D model include the insulation PBCs, electrolyte solution PBCs, and electrode deformation PBCs. The boundary conditions of Part I and Part II are shown in Figure 3a,b, respectively.

Crevice interior, Part I, boundary conditions: Because the epoxy resin is an insulator,
(2)$$\left\{\begin{array}{l} \frac{\partial E}{\partial x}=0,x=L\\ \frac{\partial E}{\partial y}=0,y=W \end{array}\right.$$

The PBC of electrolyte solution at the crevice opening is prescribed,
(3)$$E=E_{B},x=0$$
where, *E_B_* is an unknown parameter, used to match the potential of Part I with Part II in the crevice opening.

The corrosion damage *y* = *H*(*x*, *t*) on the Ni interface is due to the dissolution of Ni. The electrode deformation PBCs are determined by Ohm’s law:(4)$$k\nabla E\cdot \hat{n}=-i\left(E\right)=\frac{\partial }{\partial x}\left[\left(W+H\right)K\frac{\partial E}{\partial x}\right]$$
where, *k* denotes electrical conductivity, and $\hat{n}$ is defined as the unit normal vector to the Ni outward surface. Then, *i*(*E*) denotes the current density of the polarization profile in Figure 2. Without the damage evolution, that is *H* = 0, and assumptions of a constant current density, *i*. Formula (4) widens those applications to the damage evolution of a metal surface considering the temporal and spatial variation of current density in the gap. Due to the nonlinearity of the polarization curve *i*(*E*) and the *H*(*x*, *t*) with a free boundary, the equation displays as highly nonlinear.

Crevice exterior, Part II, boundary conditions: the boundary *x* = −*M* and *y* = −*N* + *W* of the contact between the electrolyte solution and Ni surface were much larger than the crevice width *W*, so the insulation condition could be maintained in the far field of the electrolyte solution:(5)$$\left\{\begin{array}{l} \frac{\partial E}{\partial x}=0,x=-M\\ \frac{\partial E}{\partial y}=0,y=-N+W \end{array}\right.$$

Along the boundary *x* = 0, a piecewise constant function of the electrolyte solution area is defined, 0 ≤ *y* ≤ *W*, and the exposed surface is defined by −*N* + *W* ≤ *y* < 0,
(6)$$E\left(0,y\right)=\left\{\begin{array}{l} E_{B},y\geq 0\\ E_{A},y\text{<}0 \end{array}\right.$$
where, *E_A_* is the potential value of the exposed surface of Ni metal in the electrolyte solution, and *E_B_* is the unknown potential at the opening. Since the potential on the exposed surface is artificially specified, it is not necessary to model the reduction reaction on the cathode surface.

### 2.3. Damage Equations

Assuming that the electrolyte in Part I and Part II is uniform, the potential *E* in the system is controlled by the Laplace equation:(7)$$\nabla^{2}E=0$$

The polarization curve between the current density *i*(*E*) and the potential *E* in Figure 2 may be expressed as follows [34]:(8)$$i(E)=\left\{\begin{array}{l} 1.177495\tanh\left(0.1E+12\right)+1.177505,E\leq -0.0072\\ 1.956848+0.022120x+0.000845x^{2}+0.000006x^{3},-0.072<E\leq -0.016\\ 2.046685+0.030218x+0.000804x^{2}-0.000006x^{3},-0.016<E\leq 0.108\\ -3.675890\tanh(0.08x-12.8)+3.675900,E>0.108 \end{array}\right.$$

Once *E* is determined, electrolysis was used to correlate the rate of metal dissolution, correlated with the current density by Faraday Electrolysis Law, and *i*(*E*) was used to define the space and time evolution *H*(*x*, *t*) of the corrosion damage profile in Figure 2. The damage evolution formula in Ref. [34] is
(9)$$\frac{\partial H}{\partial t}{\left(1+{\left(\frac{\partial H}{\partial x}\right)}^{2}\right)}^{-1/2}=\frac{i\left(E\right)M_{Ni}}{2F\rho_{Ni}}$$
where the left of Equation (10) represents the normal velocity of the damaged front, *M_Ni_* is the molecular weight of Ni, the exponential factor 2 in the right denominator represents the oxidation state of the species, *F* is Faraday’s constant, and *ρ_Ni_* is the density of Ni. Initially, there was no corrosion damage, so *H*(*x*, 0) = 0.
(10)$$\left\{\begin{array}{l} \frac{\partial H}{\partial t}=\frac{i(E)M_{Ni}}{2F\rho_{Ni}}\\ H(x,0)=0 \end{array}\right.$$

This is a free-boundary damage problem for the metal surface, *H*(*x*, *t*), and its shape can be derived from the solution of Equation (9). The Laplace Equation (7) in the combination Part I with Part II is restricted to the potential boundary conditions (2), (3), (5), (6), and (10).

### 2.4. Result and Discussion

#### 2.4.1. Distribution of Potential at Time Zero

The potential curves along the depth of the crevice are shown in Figure 4 As a reference, the location of the crevice opening was at position 0 on the *x*-axis and positive values are associated with increasing crevice depth towards the tip. The location of the peak current density *E_crit_* = 108 mV and a potential at which the surface is nearly passivated, *E_pass_* = 200 mV, were marked on this plot. In this paper, the position of *E_crit_* marks the minimum gap length required to initiate crevice corrosion. This position is referred to here as *x_crit_*. As expected from the *I∙R*-drop corrosion model, the width increased and the position of *E_crit_* moved deeper inside the crevice. With the width increasing, the position of *x_crit_* tended to deform in a roughly quadratic fashion (depression downward), denoting that the CPC moved into the deeper location of the crevice. Additionally reflected in Figure 4 is the relationship between *x_crit_* and the crevice width *W* under different applied potentials. Compared with Figure 4a,b, applying a larger system potential led to a larger *x_crit_* value, indicating that the higher applied potential caused the CPC to move deeper into the crevice. This is consistent with the relationship proposed by Abdulsalam [27] and Lee [33].

For time *t* = 0, the effects of different potentials on the current density and interfacial potential curves of Ni with a fixed gap *W* = 0.3 mm are shown in Figure 5. The parameters are shown in Table 1. In Figure 5a, the initiation position, as denoted by the location of *i_crit_*, was calculated to appear at approximately 2.5 mm along the crevice depth, that is, *x_crit_* = 2.5 mm. For the passive-to-active polarization data from the passive region to the active region in Figure 5b, the location of *x_crit_* was approximately 3.0 mm along the depth of the crevice, *x_crit_* = 3.0 mm. In larger gaps, the deviation from the quadratic behavior was influenced both by the area under the anode curve and the depth of the crevice. This result is expected because the onset of the higher potential must be located at a larger *L*, which is consistent with the experimental results [33].

#### 2.4.2. Distribution of Interface Potential

The potential profiles as a function of time for a 0.3 mm gap of crevice under different system potentials of 250 mV and 300 mV are shown in Figure 6. As shown on the plots, the locations of *E_crit_* and *E_pass_* from the polarization curve were marked. The potential changed quickly at the crevice opening, and the potential gradient at the opening was larger than that at the tip. Along the direction of crevice depth *L*, the change of potential tended to level off, which is consistent with experimental result [33]. With an increasing time, the location of *E_pass_* moved toward the gap opening while *E_crit_* moved away from the opening toward the gap tip. It can be explained from the following two aspects. First, as crevice damage began to develop at the location of *E_crit_*, the crevice tended to a larger width, with the result that the resistance between the opening and this location decreased. The movement of *E_crit_* to a deeper location can be reasonably explained by substituting the expression for *R* in Equation (1) into Ohm’s law. For a constant PD between the cathode and *E_crit_* and constant *I*(*i_crit_*), the *x* must change. Therefore, the PD related with *E_crit_* moved into the deeper crevice, and *x* increased. In addition, *E_pass_* moved toward the crevice opening and *E_crit_* transferred to the deeper point of crevice. The widths of these locations associated with current densities in the active peak increased. As a result, the total current near the opening of crevice increased.

As shown in Figure 7, the potential distribution for the applied potential of 250 mV and 50 h coincided with that of experimental results [25]. For the applied potential of 300 mV and 30 h, the experimental potential decreased faster than that of simulation, and compared with Lee [33] and Abdulsalam [35], it was found that all curves were consistent. However, it is also worth noting that the values of *x_crit_* for a given gap did not agree under the applied potential; higher potentials resulted in higher values of *x_crit_*, as would be expected from *I∙R* theory.

#### 2.4.3. Distribution of Current Density during Corrosion

The curves of current density with time are shown in Figure 8. The current at every point is shown as the summation of the current representing the cells on the metal surface between the deepest location and that point. The width was fixed at the gap opening (or a little loss) and the current at that point increased with time, while the voltage increased with time. In other words, applying a constant voltage *V*, an increasing in *I* is associated with a decrease in *x* by Equation (1). The damage can be predicted when crevice corrosion proceeding in two directions are away from the gap opening when the location *E_crit_* appears at the deeper location inside the crevice and toward the opening. Along the crevice depth, the total current went down. In Figure 8a, when the crevice width was *W* = 0.3 mm and the potential is *E_A_* = 250 mV, the maximum current density appeared near *x* = 2.3 mm. In Figure 8b, the maximum is near *x* = 3.0 mm under *E_A_* = 300 mV. Over time, the peak current region became wider. Two effects can explain the change in the current density distribution: the decrease of Ohmic resistance inside the crevice due to the width *W* expanding and the increase of the contact surface area between the electrode and the electrolyte due to the electrode deformation.

That the current density here is basically matched with that of the result [27], and which is shown in Figure 9. The maximum value of corrosion current density in the experiment was consistent with that of simulation; the experimental current peak occurred at about *L* = 2.1 mm, and the simulated peak occurred at about *L* = 2.3 mm.

#### 2.4.4. Damage Evolution of Ni Electrode

The damage evolution of the Ni metal surface into the crevice is shown in Figure 10. Because of the change in *E_crit_* and *E_pass_*, the damage area extended in directions from the onset point at *x* = 2.3 mm in Figure 10a and *x* = 3.0 mm in Figure 10b towards both the opening and the end of the gap. The damage deformation of the electrode surface near the crevice opening coincides with the change trend of current near the opening as mentioned above and the damage to the tip increased in line with *E_crit_*’s movement. This suggests that the shape of the current distribution in Figure 10 is not invariable. Conversely, the active-to-passive region grew, over time, to a larger area. This led to the observed sotted damage contour. Over time, the CDA expanded and moved toward the gap opening, consistent with an increase in current near the opening, and also toward the tip, in accordance with a decrease in potential.

The comparison of the damage evolution between simulation and experiment in Abdulsalam [35] is shown in Figure 11. When the potential was 250 mV, the experiment damage value was larger than that of simulation at the time of 50 h. The experiment damage threshold value was up to about 0.7 mm, and that of simulation was about 0.45 mm. At 300 mV system potential, the experiment damage was larger than that of simulation at the time of 150 h. The damage threshold value was up to about 1.1 mm, while the simulation damage threshold value was about 1.35 mm. From Figure 11, it can be seen that the fluctuation trend of experiment data and simulation data was consistent, and the corrosion point was located at about 2.3 mm on the damage section under the applied potential of 250 mV and 50 h. The experiment results were consistent with the simulation results. The CPC of experiment result appeared at about 2.7 mm along the depth at 300 mV and *t* = 150 h, and the CPC was about 3.0 mm on the simulated damage section.

#### 2.4.5. Electrode Surface Damage

The damage curves as a function of crevice gap at time equal to 50 h were calculated, and the results are shown in Figure 12. Smaller gaps were related to a narrower CDA near the crevice opening, while at the largest crevice width, the CDA was far away from the opening in order to adapt to the decrease in *I∙R* voltage drop. However, at wider crevices, the potential displayed a variation by relatively small amounts as *x*’s location from the opening gradually increased. This caused a sporadic damage area where damage is more severe than that observed for smaller width crevices. Since the effect of mass transport on gap damage is not considered in this model, the increase of damage is likely to occur only in the structure with active-to-passive transition in the electrolyte.

## 3. Simulation of X100 Crevice Corrosion

Ni element can change the crystal structure of stainless steel. The addition of Ni can form an austenite structure, improving the corrosion resistance, plasticity, weldability, and toughness of stainless steel. In Section 2 of this paper, 2-D model of the damage evolution of crevice corrosion of Ni metal in acidic medium is studied, in which simulated results were matched with the public results [27,32,33,34,35,36]. This conclusion makes it possible to perform the simulation of crevice corrosion of stainless steel. In the following, the damage behaviors of X100 steel crevice corrosion in CO_2_ solution are simulated by COMSOL 5.6. Its chemical composition is shown in Table 2.

### 3.1. Geometric Model and Boundary Conditions

The 2-D model of crevice corrosion of X100 and the meshed model are shown in Figure 13a,b, respectively, in which the crevice depth *L* was 10 mm and the crevice width *W* was taken as 0.1–0.25 mm. In general, the crevice depth *L* is considered to be much larger than the crevice width W. The electrolyte solution was a saturated CO_2_ solution of 0.5 mol/L Na_2_SO_4_ with a conductivity of 0.1 S/m. The anodic polarization curve of X100 in a saturated CO_2_ solution of Na_2_SO_4_ at 20 °C is shown in Figure 14 [31]. The X100 steel surface in contact with the electrolyte solution was considered as the anode. In this model, only the electro-migration transfer was considered while the convection transfer and electrolyte solution diffusion were ignored; the current was generated by electromigration of ions only, and the electrolytes in part I and part II were well mixed. The geometric model and boundary conditions were the same as those of Ni in Figure 3.

### 3.2. Results and Analysis

#### 3.2.1. Effect of Crevice Widths on Corrosion Damage

Crevice width is an important factor on corrosion damage; thus, here, different width of 0.1 mm, 0.15 mm, 0.2 mm, and 0.25 mm were used. Figure 15, Figure 16, Figure 17 and Figure 18 show the X100 steel electrode deformation with time at different crevice widths, where the numerical value of the legend on the right side represents the corrosion damage severity, unit μm.

Figure 19 shows that damage progressed along the gap depth, where, *x* = 1.9 mm, 2.1 mm, 2.5 mm, and 2.7 mm are from Figure 15, Figure 16, Figure 17 and Figure 18 and *x* = 3.0 mm is from Figure 10b. At the period of 0–50 weeks, the CDA of the X100 electrode surface gradually increased. As the crevice width increased, the CPC, *x_crit_* moved deeper into the gap. Only the resistivity between the gap and electrolyte met the requirement of the voltage *I∙R* > Δ*φ**, and the crevice corrosion occurred in the metal/electrolyte system at the deep interval between the gap surface *X_A/P_* and the gap ended at *x* = 10 mm. Δ*φ** is defined in the metal/electrolyte system as the difference between the potential (*E_A_* = *Ex* = 0) at the opening of crevice (*x* = 0) of the passivation zone and the critical potential (*E_x_* = *E_A/P_*) of the transition from activation zone to the passivation zone on the polarization curve. When the gap became larger, it can be seen from Equation (1) that only when the gap depth became larger could the *I∙R* > Δ*φ** condition be satisfied, resulting in corrosion.

#### 3.2.2. Damage Evolution at Time 50 Weeks

The damage profiles inside the crevice with different width at time *t* = 50 weeks are shown in Figure 20. With the gap increased, the location of the CPC gradually moved into the deeper gap. The CDA began to expand and spread in both directions from initiation corrosion point *x_crit_* toward the gap opening and the gap tip. In order to adapt to the corrosion conditions controlled by the *I∙R* drop, the CDA increased, as can be acquired from Equation (1). However, the corrosion damage threshold value basically remained unchanged, up to 0.83 mm. From damage Equation (10), applying a constant system potential, the corrosion damage maximum was related to the current density, but not to the crevice width.

## 4. Conclusions

A 2-D model of predicting the spatial-temporal damage evolution during crevice corrosion of Ni metal in sulfuric acid and X100 pipeline steel in CO_2_ well-mixed electrolyte solution was developed to simulate *I∙R* drop controlled corrosion and was consistent with some published results. Corrosion deformation, potential distribution, and current distribution on the metal surface were predicted during crevice corrosion.

(1) Under system with a constant potential, the damage of crevice corrosion of metal surface increased; when the gap increased, the critical point of corrosion inside the crevice moved into a deeper location, while the corrosion damage area gradually expanded, but the maximum corrosion damage value remained unchanged.

(2) For crevice corrosion controlled by *I∙R* voltage drop, the damage area inside the crevice expanded toward both the opening and the end of the gap. Since the potential drop in the passivation region increased with increasing current, the passivation potential point shifted towards the opening. As the crevice width increased and the electrolyte resistance decreased, the critical potential position at which the maximum crevice corrosion rate was reached moved deeper into the inner crevice.

(3) This 2-D model has the ability of predicting the spatial-temporal damage evolution that is not obtainable through the one-dimensional simplified model. The results provide a valuable explanation for variation of current density, and the crevice critical length-to-gap ratio necessary to meet the crevice corrosion controlled by *I∙R* drop theory. It is valuable to predict corrosion damage on the time and space scales.

## Figures and Tables

**Figure 1 materials-15-02329-f001:**
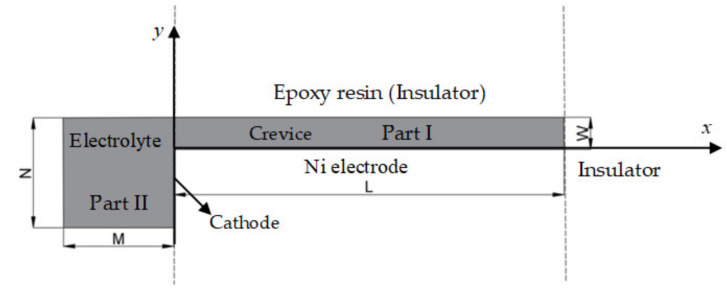
2-D electrochemical model for corrosion. The gray area is electrolyte. Not to scale.

**Figure 2 materials-15-02329-f002:**
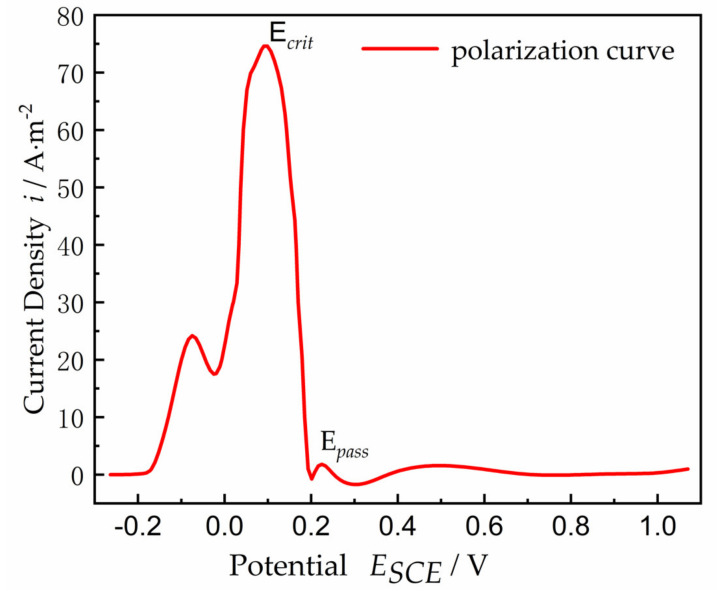
Polarization curve for Ni electrode in sulfuric acid along the active-to-passive direction from Brackman [33], indicating the critical potential, *E_crit_* = 0.108 V, and passive potential, *E_pass_* = 0.20 V.

**Figure 3 materials-15-02329-f003:**
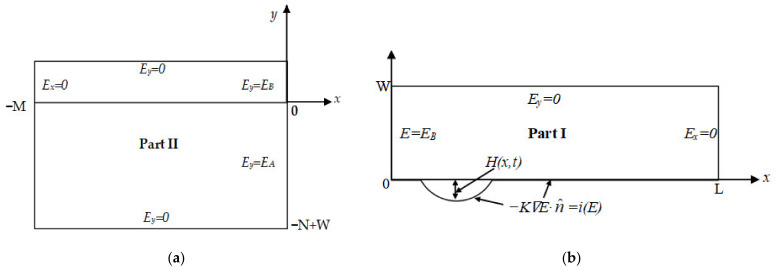
The PBCs of inside and outside the crevice: (**a**) Part I; (**b**) Part II.

**Figure 4 materials-15-02329-f004:**
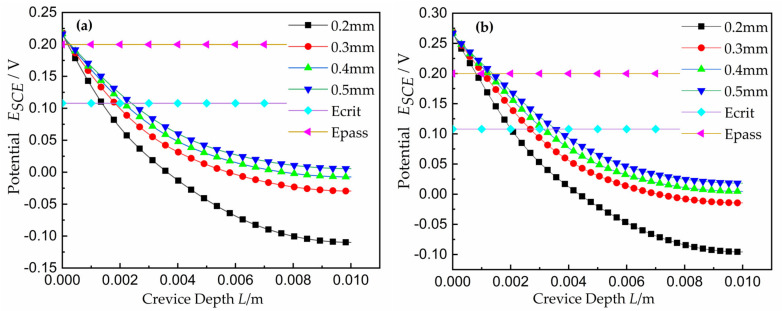
Potential curves with varying crevice width *W*, at time *t* = 0 h: (**a**) the applied potential *E_A_* = 0.25 V; (**b**) the applied potential *E_A_* = 0.30 V.

**Figure 5 materials-15-02329-f005:**
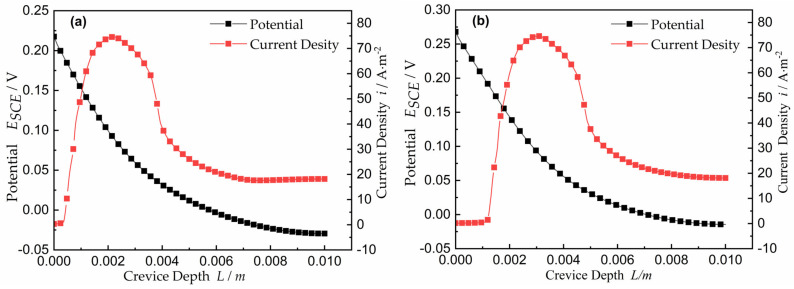
Potential E_SCE_ and current density for time *t* = 0 and *W* = 0.3 mm: (**a**) the applied potential *E_A_* = 0.25 V; (**b**) the applied potential *E_A_* = 0.30 V.

**Figure 6 materials-15-02329-f006:**
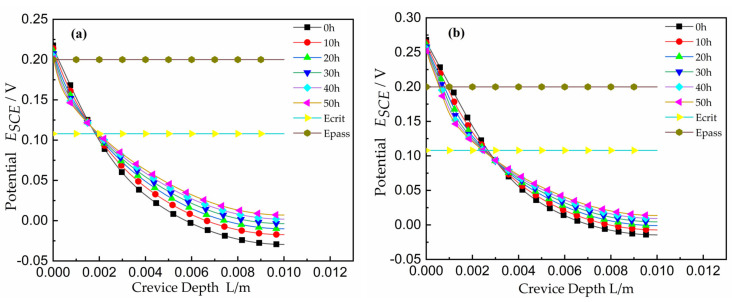
Curves of potential versus depth for the width *W* = 0.3 mm at various times: (**a**) the applied potential *E_A_* = 0.25 V; (**b**) the applied potential *E_A_* = 0.30 V.

**Figure 7 materials-15-02329-f007:**
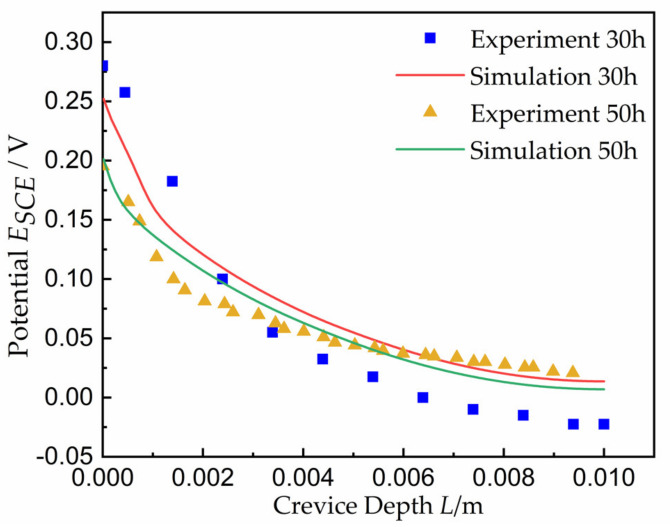
Comparison of potential with experimental results from Abdulsalam [35].

**Figure 8 materials-15-02329-f008:**
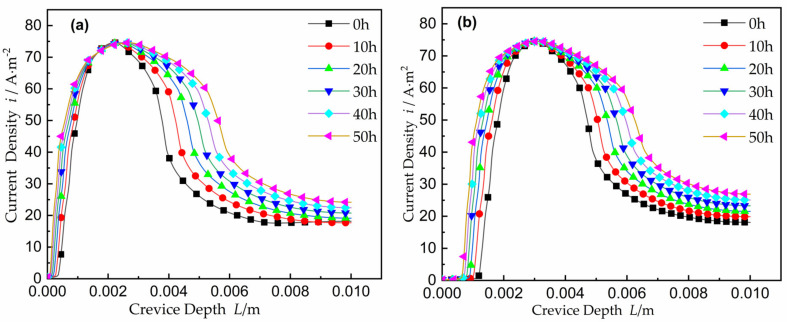
Current density with time for *W* = 0.3 mm: (**a**) the applied potential *E_A_* = 0.25 V; (**b**) the applied potential *E_A_* = 0.30 V.

**Figure 9 materials-15-02329-f009:**
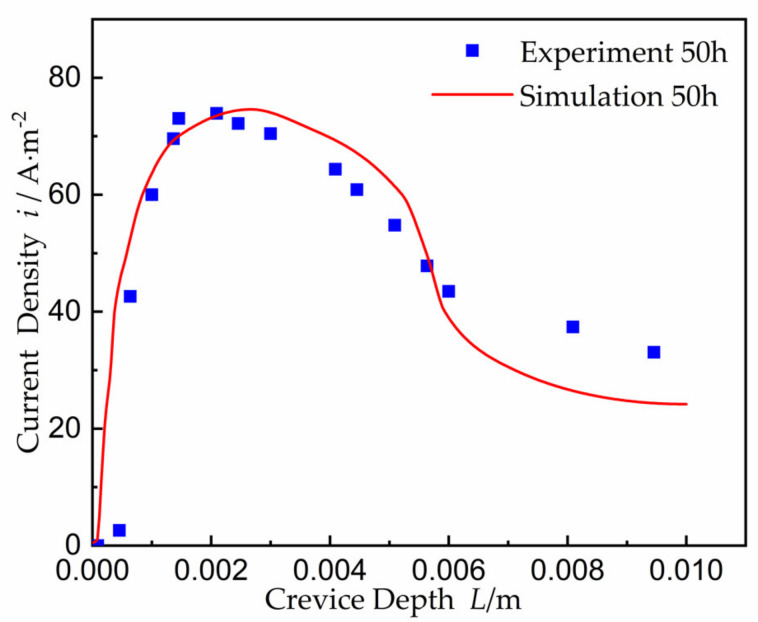
Distribution of current density at time 50 h.

**Figure 10 materials-15-02329-f010:**
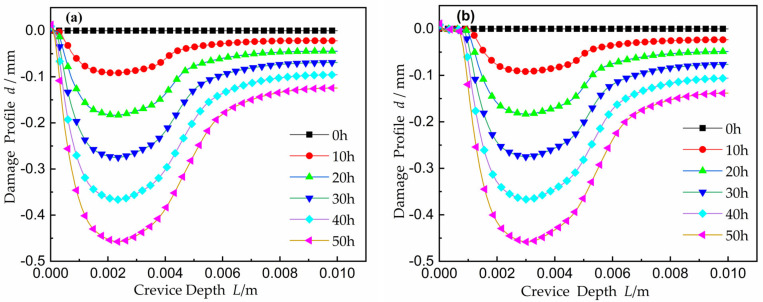
The crevice damage evolution inside the gap over time. (**a**) the applied potential *E_A_* = 0.25 V; (**b**) the applied potential *E_A_* = 0.30 V.

**Figure 11 materials-15-02329-f011:**
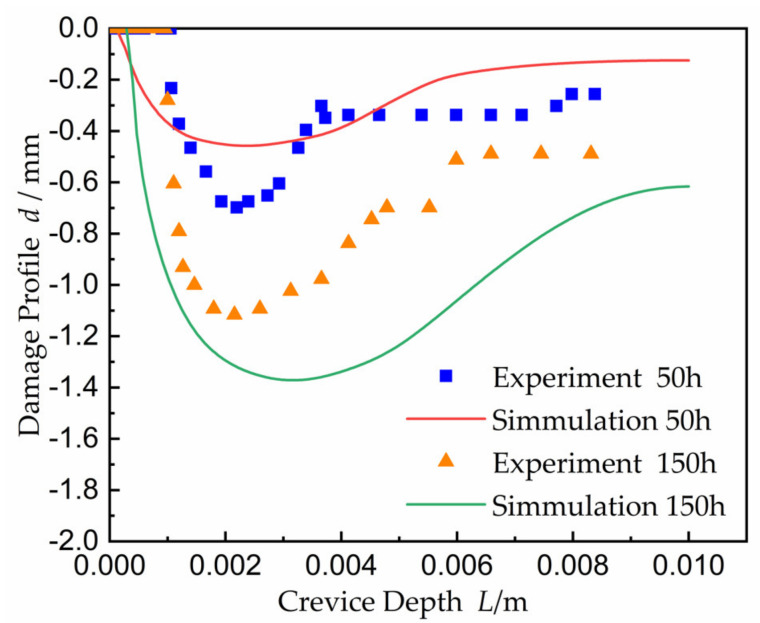
Comparison damage evolution with the result from Abdulsalam [35]. The applied potential at time 50 h is 0.25 V, and the potential of the 150 h is 0.30 V.

**Figure 12 materials-15-02329-f012:**
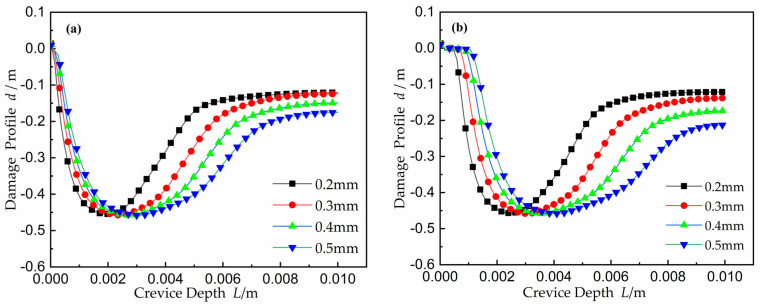
Electrode damage as a function of crevice width after 50 h: (**a**) the applied potential *E_A_* = 0.25 V; (**b**) the applied potential *E_A_* = 0.30 V.

**Figure 13 materials-15-02329-f013:**
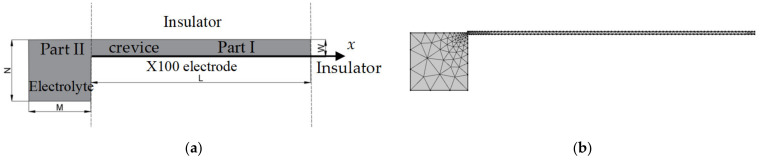
2D electrochemical model for X100 crevice corrosion: (**a**) the geometric model; (**b**) the meshed model.

**Figure 14 materials-15-02329-f014:**
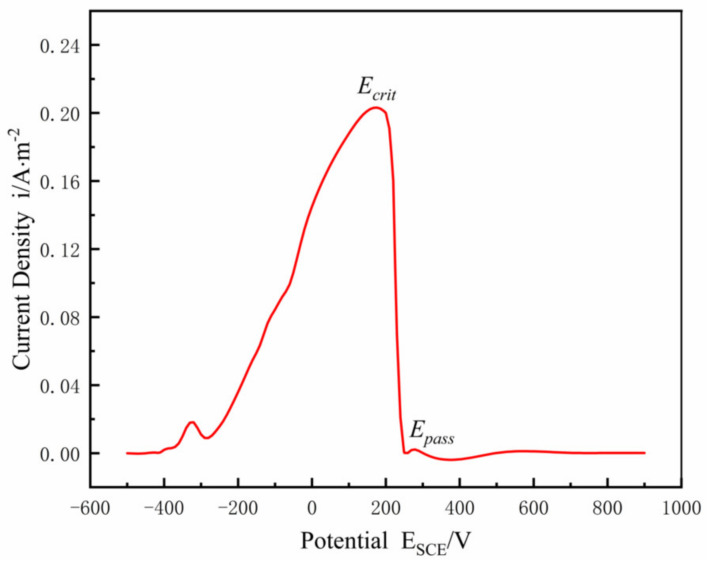
Anodic polarization curve of X100 in 0.5 mol/L Na_2_SO_4_ saturated CO_2_ solution at 20 °C.

**Figure 15 materials-15-02329-f015:**
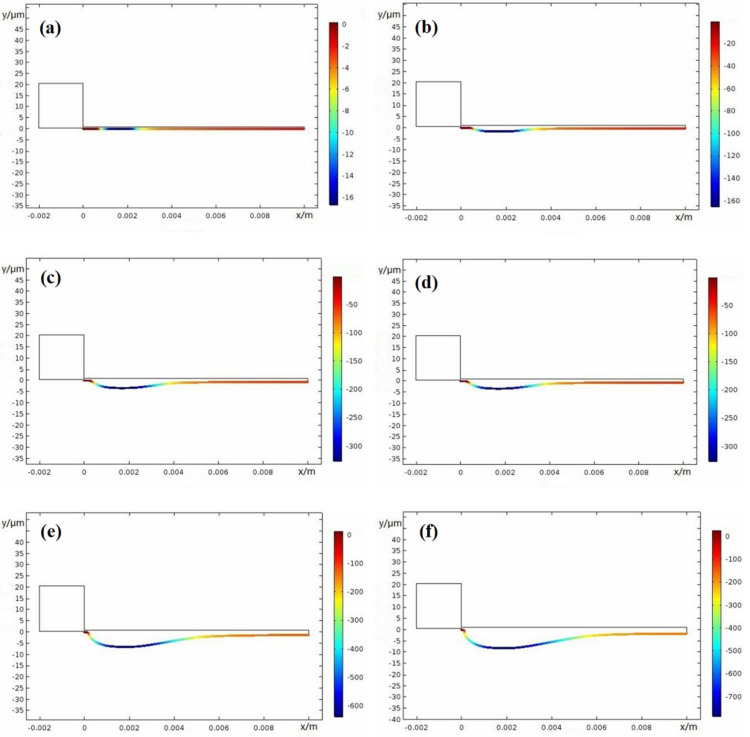
Damage evolution of X100 crevice corrosion at width *W* = 0.1 mm: (**a**) 0 weeks; (**b**) 10 weeks; (**c**) 20 weeks; (**d**) 30 weeks; (**e**) 40 weeks; (**f**) 50 weeks.

**Figure 16 materials-15-02329-f016:**
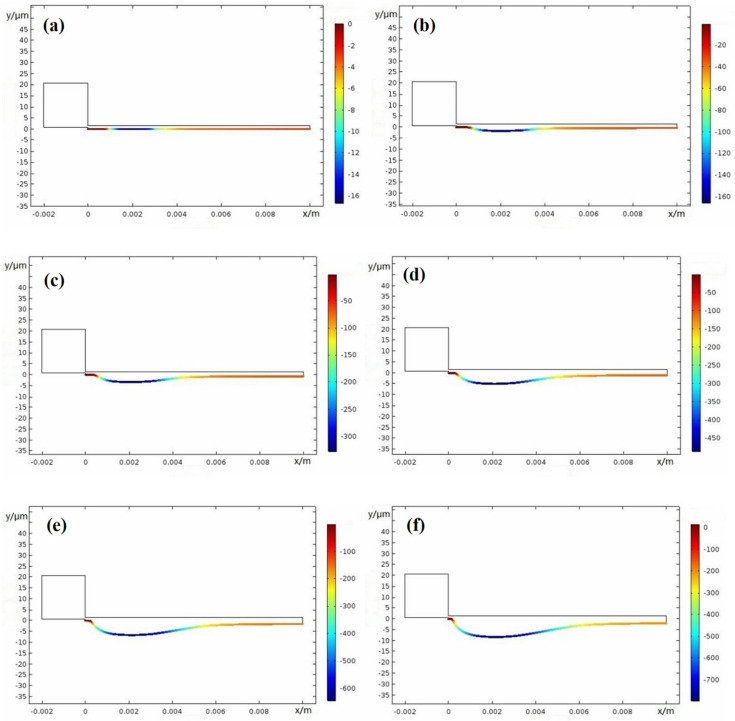
Damage evolution of X100 steel crevice corrosion at width *W* = 0.15 mm: (**a**) 0 weeks; (**b**) 10 weeks; (**c**) 20 weeks; (**d**) 30 weeks; (**e**) 40 weeks; (**f**) 50 weeks.

**Figure 17 materials-15-02329-f017:**
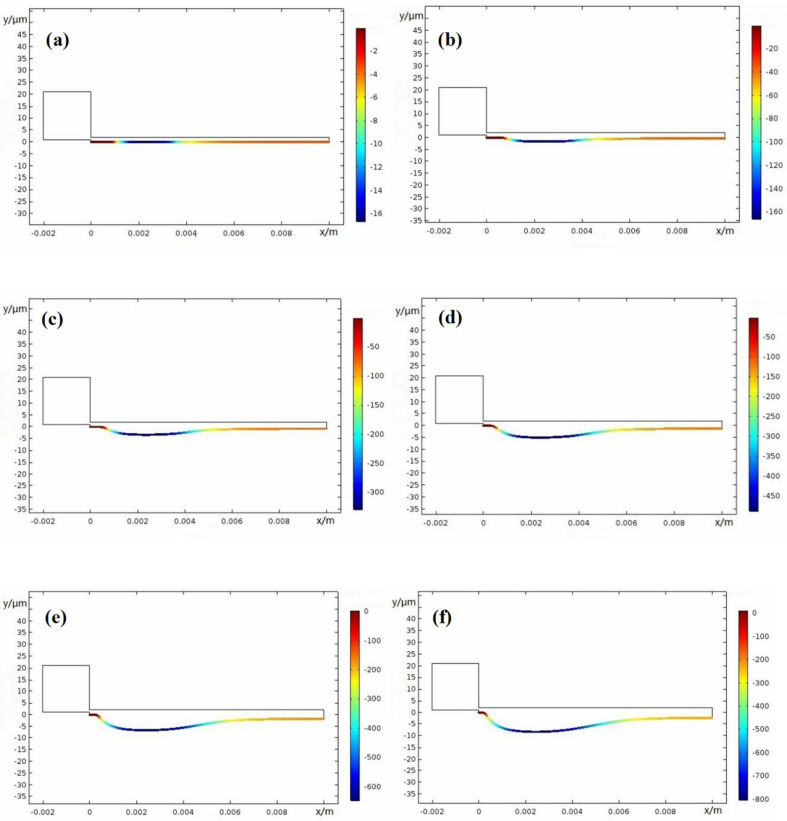
Damage evolution of X100 crevice corrosion at width *W* = 0.20 mm: (**a**) 0 weeks; (**b**) 10 weeks; (**c**) 20 weeks; (**d**) 30 weeks; (**e**) 40 weeks; (**f**) 50 weeks.

**Figure 18 materials-15-02329-f018:**
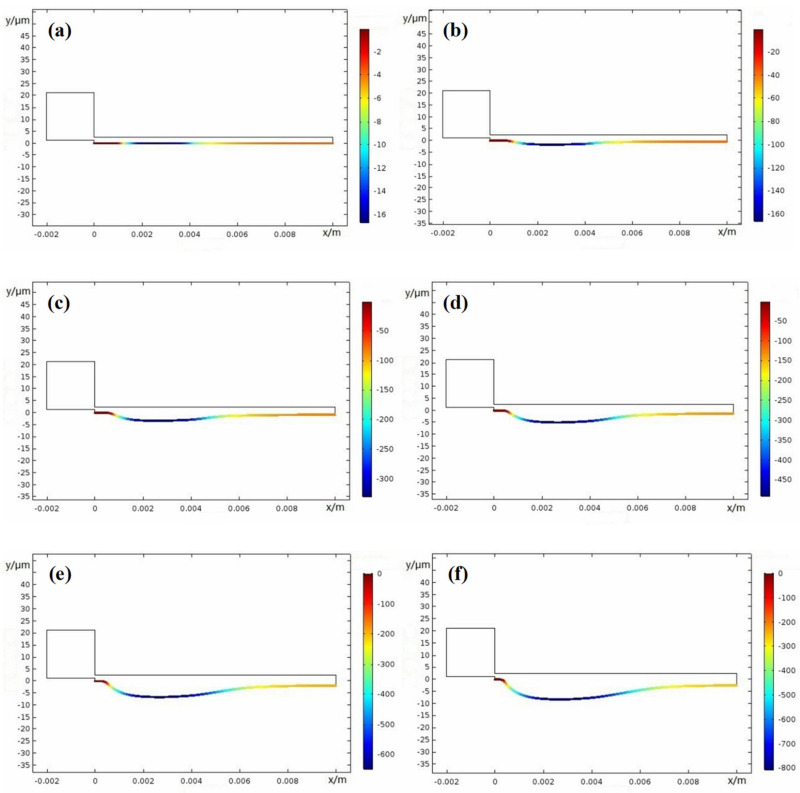
Damage evolution of X100 crevice corrosion at width *W* = 0.25 mm: (**a**) 0 weeks; (**b**) 10 weeks; (**c**) 20 weeks; (**d**) 30 weeks; (**e**) 40 weeks; (**f**) 50 weeks.

**Figure 19 materials-15-02329-f019:**
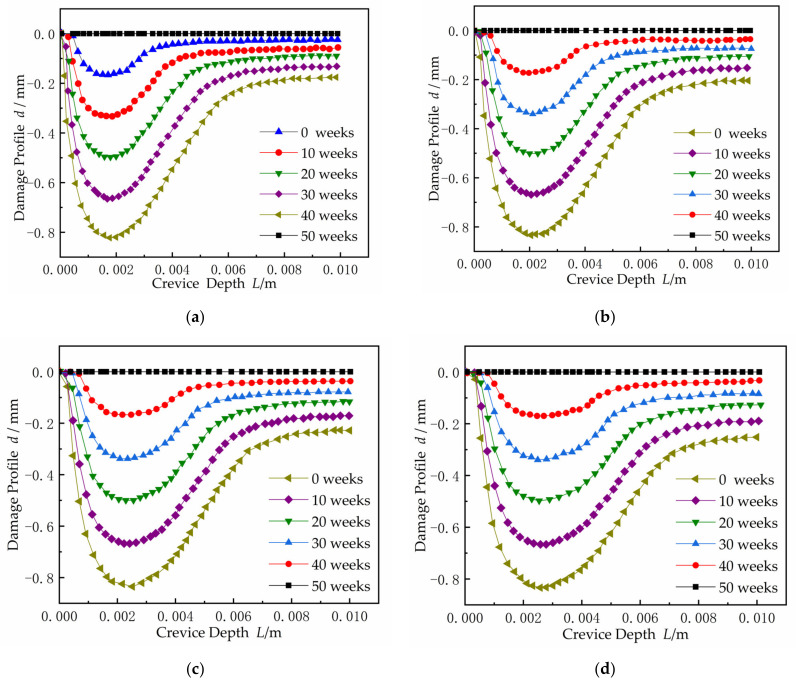
The distribution of damage curves inside the crevice over time (*d* denotes damage profile and *L* denotes crevice depth): (**a**) *W* = 0.1 mm; (**b**) *W* = 0.15 mm; (**c**) *W* = 0.2 mm; (**d**) *W* = 0.25 mm.

**Figure 20 materials-15-02329-f020:**
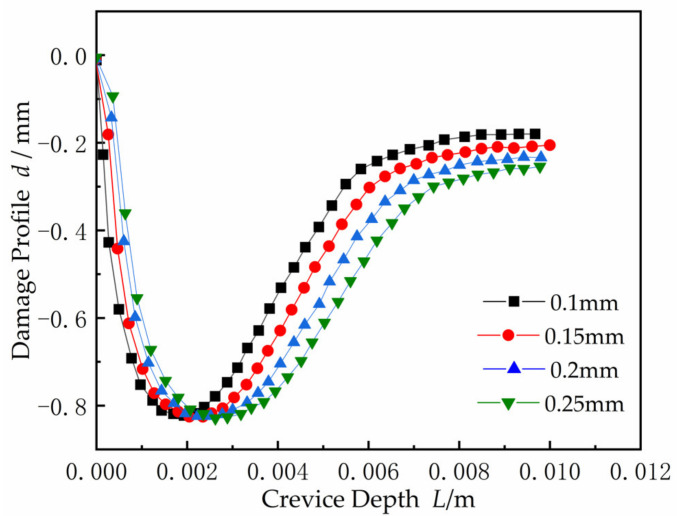
Damage profiles inside the crevice as a function of width *W* at time *t* = 50 weeks.

**Table 1 materials-15-02329-t001:** Basic parameter values.

Variable	Value	Unit
*W*	0.3	mm
*E_A_*	0.25/0.30	V
*L*	10	mm
*M*	2	mm
*N*	2	mm

**Table 2 materials-15-02329-t002:** Chemical composition of X100 pipeline steel (mass, %).

C	Si	Mn	Cr	V	Ti	Nb	Cu	Ni	Mo	Al	S	P
0.04	0.30	1.90	0.28	0.007	0.012	0.071	0.52	0.57	0.34	0.017	0.004	0.011

## Data Availability

The data used to support the findings of this study are available from the corresponding author upon request.

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
