# Peer review of "Numerical Simulation of Damage Evolution and Electrode Deformation of X100 Pipeline Steel during Crevice Corrosion"

_materials, 2022, doi:10.3390/ma15062329_

Round 1

Reviewer 1 Report

In the  paper,  2D model of predicting the spatio-temporal damage evolution during crevice  corrosion of nickel metal in sulfuric acid and X100 pipeline steel in carbon dioxide  well-mixed electrolyte solution are developed. The model allows to analyze  metal surface deformation, potential distribution and current distribution during the corrosion process. 

In my opinion the  obtained  results are  interesting. The paper is  well written  with sufficient details. Therefore, I recommend it for publication in materials. But there are some remarks on the paper:

1. In many places in the text of the paper there are no punctuation marks. Moreover, there are many typos. 

2.  Line 181: For the convenience of readers, the initial conditions for H are better to interline.

3. Line 176: It is not clear how the nonlinear with respect to H differential equation (10) was solved. The paper should have comments on this.

Reviewer 2 Report

The work is well organized and written in beautiful English.

There will be a few studies that I will suggest about corrosion and oxidation in the introduction part of the study, and I want them to be added.

If this is added to the studies I recommend, the study will be published.

Hot corrosion behavior of YSZ, Gd2Zr2O7 and YSZ/Gd2Zr2O7 thermal barrier coatings exposed to molten sulfate and vanadate salt

Comparison of calcium–magnesium-alumina-silicate (CMAS) resistance behavior of produced with electron beam physical vapor deposition (EB-PVD) method YSZ and Gd2Zr2O7/YSZ thermal barrier coatings systems

TGO growth and kinetic study of single and double-layered TBC systems

Investigation of calcium–magnesium-alumino-silicate (CMAS) resistance and hot corrosion behavior of YSZ and La2Zr2O7/YSZ thermal barrier coatings (TBCs) produced with CGDS method

Oxidation and hot corrosion resistance of HVOF/EB-PVD thermal barrier coating system

Investigation of the effect of V2O5 and Na2SO4 melted salts on thermal barrier coatings under cyclic conditions

Evaluation of oxidation and thermal cyclic behavior of YSZ, Gd2Zr2O7 and YSZ/Gd2Zr2O7 TBCs

Isothermal oxidation and thermal cyclic behaviors of YSZ and double-layered YSZ/La2Zr2O7 thermal barrier coatings (TBCs)

Evaluation of Hot Corrosion Behavior of APS and HVOF Sprayed Thermal Barrier Coatings (TBCs) Exposed

Reviewer 3 Report

The paper is well written I have some minor comments:

(1) The paper needs nomenclature to reduce the number of pages.

(2) Can the author explain how they do the contact in COMSOL software?

(3) Can the author add the COMSOL software?

(4) Do the authors use finite element method if yes can the authors add convergence test?

(5) The font size in the figures is smaller than the text in the manuscript. Please unify that and the increase the resolution of these figures.

Round 2

Reviewer 2 Report

The work is eligible for publication

Reviewer 3 Report

The paper is well written I have some minor comments:

(1) The paper needs nomenclature to reduce the number of pages.

(2) Can the author explain how they do the contact in COMSOL software?

(3) Can the author add the COMSOL software?

(4) Do the authors use finite element method if yes can the authors add convergence test?

(5) The font size in the figures is smaller than the text in the manuscript. Please unify that and the increase the resolution of these figures.

Round 3

Reviewer 3 Report

I have two points:

(1) I asked the author to do convergence test of finite element method between the degree of freedom against number of total degrees of freedom but I did not see it in the new version of the manuscript.

(2) I asked the authors to explain the steps of finite element method using COMSOL software but I did not see it in the new version of the manuscript.
